# Risk Stratification for Pacemaker Implantation after Transcatheter Aortic Valve Implantation in Patients with Right Bundle Branch Block

**DOI:** 10.3390/jcm11195580

**Published:** 2022-09-22

**Authors:** Simon Schoechlin, Martin Eichenlaub, Björn Müller-Edenborn, Franz-Josef Neumann, Thomas Arentz, Dirk Westermann, Amir Jadidi

**Affiliations:** Division of Cardiology & Angiology, University Heart Center Freiburg-Bad Krozingen, University Freiburg, Südring 15, 79189 Bad Krozingen, Germany

**Keywords:** TAVI, permanent pacemaker implantation, right bundle branch block

## Abstract

Background: Permanent pacemaker implantation (PPI) after transcatheter valve implantation (TAVI) is a common complication. Pre-existing right bundle branch block (RBBB) is a strong risk factor for PPI after TAVI. However, a patient-specific approach for risk stratification in this subgroup has not yet been established. Methods: We investigated TAVI patients with pre-existing RBBB to stratify risk factors for PPI and 1-year-mortality by detailed analysis of ECG data, RBBB morphology and degree of calcification in the implantation area assessed by computed tomography angiography. Results: Between 2010 and 2018, 2129 patients underwent TAVI at our institution. Among these, 98 pacemaker-naïve patients with pre-existing RBBB underwent a TAVI procedure. PPI, because of relevant conduction disturbances (CD), was necessary in 43 (43.9%) patients. PPI was more frequently indicated in women vs. men (62.1% vs. 32.8%, *p* = 0.004) and in men treated with a self-expandable vs. a balloon-expandable valve (58.3% vs. 26.5%, *p* = 0.035). ECG data (heart rhythm, PQ, QRS, QT) and RBBB morphology had no influence on PPI rate, whereas risk for PPI increased with the degree of calcification in the left septal His-/left bundle branch-area to a 9.375-fold odds for the 3rd tertile of calcification (1.639–53.621; *p* = 0.012). Overall, 1-year-mortality was comparable among patients with or without PPI (14.0% vs. 16.4%; *p* = 0.697). Conclusions: Patients with RBBB undergoing TAVI have a high risk of PPI. Among this subgroup, female patients, male patients treated with self-expandable valve types, patients with high load/degree of non-coronary LVOT calcification and patients with atrial fibrillation need enhanced surveillance for CD after procedure.

## 1. Introduction

Conduction disturbances that lead to permanent pacemaker implantation (PPI) after transaortic valve implantation (TAVI) are a common complication. Nevertheless, the indication for PPI is still an area of uncertainty [1,2,3,4,5,6]. The phenomenon of new onset left bundle branch block (LBBB) gained attention in several scientific reports [7,8,9,10,11,12,13], as it has been reported to be associated with increased all-cause and cardiovascular mortality and an increased necessity of subsequent PPI [11,14]. 

In nearly every scientific report, a pre-existing RBBB was associated with higher rates of PPI and is nowadays an established independent risk factor for PPI after TAVI, besides other risk factors as depth of valve positioning, oversizing or LVOT/annulus ratio, valve type and calcification [15,16,17,18,19,20].

Yet, literature lacks a differentiated analysis of patients with pre-existing RBBB, as there are no data on a further subclassification of patients RBBB according to ECG criteria and RBBB morphology in relation to the other risk factors. Conduction disturbances with proximity closer to the HIS-Bundle should lead to a more vulnerable conduction and may be easier affected by an implanted valve; this may be detected by a different RBBB morphology. Therefore, using the large TAVI cohort at our institution, we searched for novel ECG, rhythm or anatomical predictors for the identification of patients with underlying RBBB who are at high risk for PPI or present an increased 1-year cardiac mortality if no PPI was performed after TAVI.

We wanted to investigate the influence of RBBB including RBBB morphology after TAVI. Therefore, we included risk factors such as detailed ECG criteria, RBBB morphology and calcification in the area of valve implantation can lead to a better risk stratification for PPI and prediction of 1-year-mortality in patients undergoing TAVI with pre-existing RBBB. Our hypothesis was that RBBB morphology has an influence on PPI rates and higher amount of calcification in the implantation area leads to more PPI.

## 2. Materials and Methods

### 2.1. Study Population

Our single centre, retrospective, observational study, was based on a total of 2129 consecutive patients who underwent a transfemoral TAVI procedure at our institution from 01/2010 to 12/2018. After exclusion of 117 patients with already-implanted pacemaker and exclusion of patients without bundle branch block, patients with left bundle branch block and patients with bifascicular block (left anterior hemiblock + right bundle branch block), a total of 98 patients with isolated complete RBBB (QRS ≥ 120 ms) prior to TAVI remained for further analysis. Clinical characteristics and procedural data are displayed in Table 1 and Table 2. Patients with aortic stenosis were primarily assessed by transthoracic echocardiography; systolic annular dimensions and grade of calcification were obtained by computed tomography angiography (CTA), as previously described [21]. The multidisciplinary institutional heart team assessed patients for eligibility, procedural feasibility, access route, valve type and size. TAVI patients received an electrocardiogram on admission, immediately after procedure, and on days 2 and 3 after TAVI and before discharge. The diagnosis of LBBB and AV block followed known recommendations [22].

### 2.2. Follow-Up and Endpoints

As part of our routine quality assurance programme, TAVI-patients were contacted by questionnaire and standardized telephone calls at 30 days, 6 months and 1 year after procedure. We report a complete follow-up for this cohort over 1 year, with no patients lost in follow-up. Primary endpoint was pacemaker implantation until discharge, secondary endpoint was 1-year all-cause mortality. The Institutional Clinical Research and Ethics Committee approved the study, and all patients gave written informed consent.

### 2.3. CTA and Calcification

The contrast-enhanced CT examinations were performed on a second-generation dual-source CT scanner (Somatom Definition Flash, Siemens Healthcare, Forchheim, Germany). The scan protocol contained a retrospective ECG-gated data acquisition of the aortic root and a non-gated aorto- femoral high-pitch spiral dual-source acquisition. The dual phasic injection protocol with a total of 50–70 mL iodinated contrast agent (Imeron 400, Bracco, Konstanz, Germany) consisted of an initial bolus at 4 mL/s followed by a 50:50% mixture with NaCl at 4 mL/s. Image acquisition was started 7 s after the region of interested, placed in the left atrium, and reached 70 Hounsfield units (bolus tracking technique). 

For the purpose of this study, only the angiography extending from upper chest to both access routes of the femoral arteries was evaluated with dedicated post-processing software (3mensio V10, Pie Medical Imaging, Maastricht, The Netherlands).

For quantification of aortic valve calcifications, mean Hounsfield units (HU) and standard deviations of HUs (SD) in contrast enhanced images were measured using a region of interest (ROI) including two thirds of the cross-section area of the ascending aorta 3 cm above the aortic valve. Second, aortic valve calcification was quantified using individual HU cut-offs 4 SD above the mean value of the ROI. The aortic wall, the coronary arteries and the mitral valve annulus were excluded from the analysis. LVOT calcifications were quantified using same HU cut-off. Whole LVOT was defined as 10 mm and upper LVOT 2 mm below the aortic valve annulus. These measurements were performed by two blinded, independent radiologists at our centre with no affiliation to this study project.

### 2.4. Monitoring after TAVI, ECG and PPI

Monitoring after TAVI consisted of telemetry on intensive care unit (ICU). Temporary transvenous pacemaker for 24 h established protection of conductions disturbances (CD). Indications for PPI in patients with RBBB are displayed in Appendix A. We analysed ECG prior procedure for heart rhythm, durations of PQ, QRS, QT, QTc and morphological QRS patterns: rsR’, RSR’, rSR’, Rsr’, Rsr’ (Figure 1). Patients with ejection fraction <40% and expected high proportion of ventricular stimulation received systems for cardiac resynchronization. 

### 2.5. Statistical Analysis

Statistical analysis was performed with SPSS v. 25.0 statistical software (IBM, New York, NY, USA). Continuous variables are presented as mean ± SD and categorical variables as frequencies and percentages. Continuous variables were tested with Kolmogorov–Smirnov-Test for normal distribution. For normally distributed continuous variables comparing 2 groups, we used Student’s *t* test. For non-normally distributed continuous variables, we used Mann–Whitney-U test comparing 2 groups. Testing of categorical variables was performed by χ^2^-test. 

Survival was assessed according to the Kaplan–Meier method and compared by log-rank test. We performed a logistic regression model to calculate odds ratios with associated 95% confidence intervals. For multivariable regression analysis, all variables with statistical significance levels at *p* < 0.1 were included (sex, left ventricular ejection fraction, valve type). Statistical significance was assumed when null hypothesis could be rejected at *p* < 0.05. 

## 3. Results

### 3.1. Study Population

Clinical characteristics are displayed in Table 1. Mean age was 82.2 years; 62.2% were male and the mean logistic EuroSCORE was 19.6 ± 16.5%. Patients, who needed PPI were less likely male (46.5% vs. 74.5%, *p* = 0.004) and had significantly better left ventricular ejection fraction (56 ± 10 vs. 51 ± 11, *p* = 0.016). 

Procedural characteristics are displayed in Table 2. Implanted valve types were CoreValve, Evolute R, Symetis and Evolute Pro (summarized as self-expandable valves) and Sapien XT and Sapien 3 (summarized as balloon-expandable valves). Patients who needed PPI received numerically more often self-expandable valve types (32.6% vs. 16.4%, *p* = 0.060). No valve-in-valve procedures were performed in patients, who needed PPI after TAVI vs. 3 (5.5%) patients, who did not need PPI. 

ECG characteristics are displayed in Table 3. In total, 74% of the patients were in sinus rhythm and the remaining 26% percent of patients were in atrial fibrillation or atrial flutter. There were no significant differences in the ECG characteristics or RBBB morphologies of patients who needed PPI vs. those who did not. 

### 3.2. Pacemaker Implantation

The overall rate of PPI in patients with RBBB was 43.9% until discharge (see indications for PPI, Appendix A); one more patient received a pacemaker after discharge on day 35 after procedure. The vast majority of PPI were indicated because of third-degree AV block (79.1%), followed by symptomatic bradycardia (14.0%), alternating bundle branch block (4.6%), and high degree (2.3%). Time of PPI was mainly on the day of the TAVI-procedure and the first day after TAVI-procedure (day 0 = 55.8%, day 1 = 27.9%, day 2 = 4.7%, later than day 3 = 11.6%). 

Patients who needed PPI were significantly more often female and received self-expandable valves (Table 4). While female patients showed no difference with regard to the implanted valve type on their risk for PPI, male patients had a significantly higher risk for PPI, when they underwent a self-expandable valve type (58.3% vs. 26.5%, *p* = 0.035).

In the univariable regression analysis, HR (95% CI) for PPI in patients with RBBB was HR 0.297 (95% CI 0.127–0.697; *p* = 0.005) for males. For self-expandable valve types, the HR (95% CI) for PPI was HR 2.467 (95% CI 0.947–6.429; *p* = 0.065). 

In multivariable analysis, female sex was the strongest risk factor (HR 4.431, 95% CI 1.609–12.203; *p* = 0.004), followed by self-expandable valve type (HR 3.877, 95% CI 1.045–14.386; *p* = 0.043) with a *p* for interaction of 0.206. Left ventricular ejection fraction was not an independent risk factor in multivariable regression analysis (HR 1.031, 95% CI 0.983–1.080; *p* = 0.210).

RBBB morphology was not associated with PPI, as well as other ECG characteristics (Appendix A).

### 3.3. Calcification at the Non-Coronary Cusp Left Ventricular Outflow Tract

The volume of calcification (measured in mm^3^) for different regions of the LVOT was analysed at CTA and is displayed in Appendix A. CTA data were available for 53 patients. Quantitative analysis of LVOT calcification revealed a significantly higher degree of calcification in the area of NCC-LVOT in patients who needed PPI (94 ± 141 mm^3^ vs. 52 ± 161, *p* = 0.036). In contrast, the volume of calcification in LVOT areas distant from the proximal left conduction system (i.e., near the left or the right aortic cusps) did not differ between patients with vs. without need for PPI. Similarly, the volume of calcification of the aortic valve and the entire upper LVOT did not differ significantly between patients with vs. without PPI.

For further analyses, tertiles for the grade of calcification were created (Appendix A). Again, the highest degree of calcification, represented by the 3rd tertile, was found in patients who needed PPI (*p* = 0.022), whereas calcification in different regions had no influence on PPI. Logistic regression analysis for tertiles of calcification calculated a 9.375-fold odds for PPI for the 3rd tertile (1.639–53.621; *p* = 0.012), and even for the 2nd tertile a 6.0-fold risk for PPI (1.049–34.317; *p* = 0.044)

### 3.4. Mortality

During the 1-year follow-up of our cohort (*n* = 98), overall mortality was 15.3%. PPI did not affect 1-year mortality (14.0% vs. 16.4%, *p* = 0.697), Figure 2. RBBB morphology had no significant influence on 1-year mortality (Appendix A).

## 4. Discussion

Our hypothesis that there are additional factors for a better risk stratification in patients with pre-existing RBBB for PPI after TAVI can be partially confirmed: female sex is the strongest risk factor in patients for PPI, a high amount of LVOT/non-coronary cusp calcification is associated with a higher risk for PPI, and male patients with balloon-expandable valve types have the lowest risk for PPI. Interestingly, ECG criteria, including the different RBBB morphologies, did not help to further stratify risk for PPI in patients with pre-existing RBBB. Patients with pre-existing RBBB and atrial fibrillation who are not treated with PPI have an increased mortality risk of 35.7% at one-year following TAVI. As autopsies were not performed, we do not know causes of death.

### 4.1. RBBB and Mortality

We can confirm previous findings that patients with RBBB are at the same risk for death after TAVI as compared to no-RBBB [23]. 

### 4.2. Conduction Disturbances and Pacemaker

An interesting finding was that male patients who received self-expandable valves are nearly at the same increased risk for PPI as female patients, in whom the choice of valve type did not affect their risk for PPI. The reason for the higher number of PPI in female patients is unclear, and we think, that the main factor for our finding in male patients, treated with balloon-expandable valve types, is the male sex. A possible reason could be a relevant oversizing of valves, as the aortic annulus in female individuals tends to be smaller in respect to their body size. Oversizing is a known and relevant risk factor for PPI^19^ and could also be an independent risk factor in patients with RBBB. As women tend to be smaller in body size, they also show smaller aortic annulus (Appendix A). Nevertheless, the heart team discusses anatomical measurements of the aortic annulus in order to choose the best fitting valve. Therefore, we calculated the oversizing factors in respect to patients’ sex. Oversizing was calculated by setting valve sizes in relation to the annulus diameter, so a smaller annulus in women could be excluded as a relevant factor. In our analyses, there were no significant differences among sexes (1.08 ± 0.08 vs. 1.11 ± 0.09; *p* = 0.117) in respect to their need for a pacemaker. This could be caused by the limited sample size and should be evaluated in a bigger cohort.

### 4.3. Conduction Disturbances and Calcification of the LVOT-Non-Coronary Cusp Junction

We report that calcification in the area of LVOT/non-coronary cusp is significantly associated with risk for PPI after TAVI in patients with RBBB. Logistic regression analysis for tertiles of calcification calculated a 9.375-fold odds for PPI for the 3rd tertile (1.639–53.621; *p* = 0.012). The anatomical area corresponding to the left His-Bundle and proximal left bundle branch is located at the membranous portion of the interventricular septum, adjacent to the non-coronary cusp area of the left ventricular outflow tract (NCC-LVOT). Therefore, an explanation for higher rates of PPI in patients with calcification in this area could be the anatomical proximity to the proximal left ventricular conduction system. Here, calcification of the interventricular membranous septum and/or the non-coronary LVOT area might compress structures of the proximal left ventricular conduction system during/following valve expansion/deployment. We analysed two exemplary CT scans of patients with a high amount of calcification in the LVOT/non-coronary cusp that reveal further explanations: one is that calcified spot are directly shifted into the septum (Appendix A). The other, where the highest LVOT/non-coronary cusp calcification was detected on the opposite direction of the LVOT in the area of the mitral valve/ring, might lead to a high radial force of the expanding valve towards the septum, where the calcification might act as an abutment (Appendix A).

### 4.4. Risk Stratification in Patients with RBBB

When planning TAVI, patients with RBBB, and the risk factors described here, should be monitored carefully with regard to conduction disturbances and need for PPI after TAVI. Female patients, male patients receiving self-expandable valves and patients with calcified LVOT/non-coronary cusp are at high risk for PPI. Finally, patients with RBBB and atrial fibrillation a strategy of electrical cardioversion to sinus rhythm might be considered that allows detecting and categorizing atrio-ventricular conduction disturbances with higher sensitivity than during ongoing AF. Alternatively, invasive electrophysiological study may be considered to determine the atrio-ventricular and His-ventricular (HV) conduction times for the risk stratification for PPI. 

## 5. Limitations

Its retrospective design needs to be considered as a main limitation of this study. To the best of our knowledge, this is the largest study focusing on the impact of right bundle branch blocks on PPI and survival. However, the limited number of observed events restricts detection of small differences in outcome as well as our ability to adjust for confounders. Pacemaker surveillance was performed by the referring cardiologists in most patients. Therefore, we could not analyse pacemaker data systematically during follow-up. Thus, we were unable to assess the proportion of patients with PPI who never needed this device. Another limitation is that quantification of calcification by CTA was only available in 53 patients (54.1%).

## 6. Conclusions

Patients with right bundle branch block are at high risk for pacemaker implantation after TAVI. While right bundle branch block morphologies do not affect the risk for pacemaker implantation, female sex, male individuals treated with self-expandable valve types and calcification of LVOT/non-coronary cusp (left His- and left bundle branch area) are the characteristics that are associated with increased risk for development of conduction disturbances post TAVI. These patients need careful monitoring/surveillance, and the heart team should discuss which patients could profit from PPI prior to procedure by respecting valve design, calcification and sex. Moreover, valve type selection may reduce pacemaker implantation rates in male patients. Patients with RBBB and atrial fibrillation might profit from electrophysiological study or a liberal PPI strategy to reduce mortality following TAVI.

## Figures and Tables

**Figure 1 jcm-11-05580-f001:**
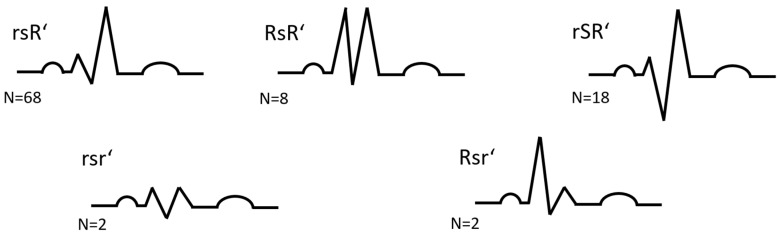
RBBB morphologies, lead V1 (RBBB = right bundle branch block).

**Figure 2 jcm-11-05580-f002:**
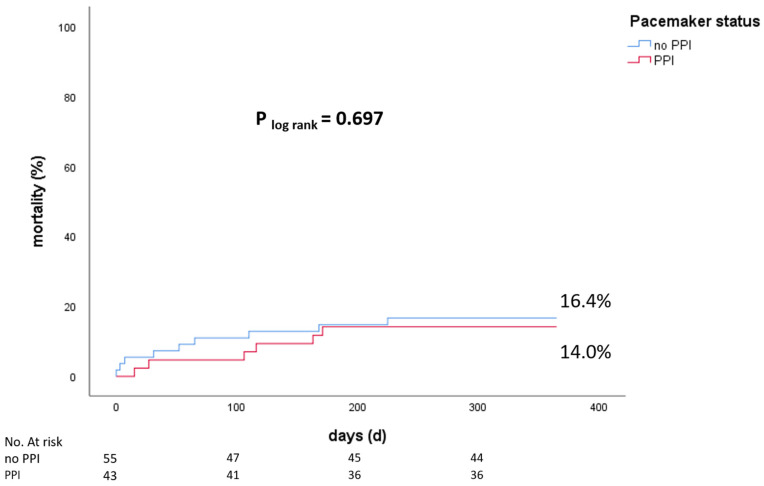
1-year mortality of patients with PPI and without PPI (PPI = permanent pacemaker implantation).

**Table 1 jcm-11-05580-t001:** Clinical characteristics of patients with pre-existing RBBB.

	All Patients (*n* = 98)	PPI after TAVI (*n* = 43)	No PPI after TAVI (*n* = 55)	*p* Value
Age, yrs	82.2 ± 6.3	81.3 ± 7.5	83.0 ± 5.3	0.832
Male	61 (62.2)	20 (46.5)	41 (74.5)	0.004
Logistic EuroSCORE, %	19.6 ± 16.5	18.6 ± 14.8	20.3 ± 17.8	0.728
BMI	27.2 ± 5.2	27.6 ± 5.8	26.9 ± 4.7	0.487
Mean aortic gradient, mm Hg	46 ± 14	44 ± 14	48 ± 14	0.140
LVEF, %	54 ± 11	56 ± 10	51 ± 11	0.016
Hypertension	89 (90.8)	40 (93)	49 (89.1)	0.504
Dyslipidaemia	73 (74.5)	33 (76.7)	40 (72.7)	0.651
Diabetes	32 (32.7)	15 (34.9)	17 (30.9)	0.677
Glomerular filtration rate	54.6 ± 24.4	57 ± 27.7	52.9 ± 21.8	0.766
Coronary artery disease	65 (66.3)	30 (69.8)	35 (63.6)	0.524
Peripheral Artery Disease	15 (15.3)	7 (16.3)	8 (14.5)	0.813
Cerebrovascular Disease	19 (19.4)	9 (20.9)	10 (18.2)	0.733
Pulmonal Hypertension	46 (46.9)	21 (48.8)	25 (45.5)	0.739
Previous Myocardial Infarction	12 (12.2)	5 (11.6)	7 (12.7)	0.869
Previous CABG	12 (12.2)	5 (11.6)	7 (12.7)	0.869
Previous Aortic Valve Surgery	3 (3.1)	-	3 (5.5)	0.120

Values are mean ± or n (%). BMI = body mass index, CABG = coronary artery bypass graft; Euro SCORE = European System for Cardiac Operative Risk Evaluation; LVEF = left ventricular ejection fraction, PPI = permanent pacemaker implantation, RBBB = right bundle branch block, TAVI = transaortic valve implantation.

**Table 2 jcm-11-05580-t002:** Procedural characteristics of patients with pre-existing RBBB.

	All Patients (*n* = 98)	PPI after TAVI (*n* = 43)	No PPI after TAVI (*n* = 55)	*p* Value
Dilatation				
Pre-dilation	18 (18.4)	6 (14)	12 (21.8)	0.318
Post-dilatation	19 (19.4)	8 (18.6)	11 (20)	0.862
Valve Type				0.060
Self-expandable	23 (23.5)	14 (32.6)	9 (16.4)	
Balloon-expandable	75 (76.5)	29 (67.4)	46 (83.6)	
Valve Size				0.738
23 mm	24 (24.5)	10 (23.2)	14 (25.5)	
26 mm	46 (46.9)	20 (46.5)	26 (47.3)	
29 mm	28 (28.6)	13 (30.2)	15 (27.3)	
Valve in Valve	3 (3.1)	-	3 (5.5)	0.120

Values are mean ± or *n* (%). PPI = permanent pacemaker implantation, RBBB = right bundle branch block, TAVI = transaortic valve implantation.

**Table 3 jcm-11-05580-t003:** Baseline ECG characteristics.

	All Patients (*n* = 98)	PPI after TAVI (*n* = 43)	No PPI after TAVI (*n* = 55)	*p* Value
Sinus rhythm	72 (73.5)	31 (72.1)	41 (74.5)	0.785
Atrial Fibrillation/Flutter	26 (26.5)	12 (28.9)	14 (25.5)	0.785
Heart rate	71 ± 10	75 ± 13	73 ± 12	0.732
PQ duration (ms)	189 ± 48	191 ± 53	188 ± 45	0.820
QRS duration (ms)	141 ± 15	142 ± 18	140 ± 12	0.940
QT duration (ms)	424 ± 36	410 ± 36	437 ± 32	0.405
QTc duration (ms)	465 ± 27	457 ± 26	473 ± 26	0.987
RBBB morphology				0.771
rsR’	68 (69.4)	31 (72.1)	37 (67.3)	
RsR’	8 (8.2)	3 (7)	5 (9.1)	
rSR’	18 (18.4)	8 (18.6)	10 (18.2)	
Rsr’	2 (2.0)	-	2 (3.6)	
Rsr’	2 (2.0)	1 (2.3)	1 (1.8)	
r-s-duration (ms)	46 ± 15	46 ± 17	45 ± 12	0.701
s-r’-duration (ms)	98 ± 15	99 ± 15	97 ± 14	0.414

Values are mean ± or *n* (%). ECG = electrocardiogram, PPI = permanent pacemaker implantation, RBBB = right bundle branch block, TAVI = transaortic valve implantation.

**Table 4 jcm-11-05580-t004:** Distribution of PPI according to valve type and sex (*p* for interaction = 0.206). (PPI = permanent pacemaker implantation, TAVI = transaortic valve implantation).

	PPI after TAVI (*n* = 43)	No PPI after TAVI (*n* = 55)	*p* Value
Male with self-expandable (*n* = 12)	7 (58.3)	5 (41.7)	
Male with balloon-expandable (*n* = 49)	13 (26.5)	36 (73.5)	
Female with self-expandable (*n* = 11)	7 (63.6)	4 (36.4)	
Female with balloon-expandable (*n* = 26)	16 (61.5)	10 (38.5)	

## Data Availability

Data, supporting results of this study, can be assessed on request to the corresponding author.

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
