# Peer review of "Risk Stratification for Pacemaker Implantation after Transcatheter Aortic Valve Implantation in Patients with Right Bundle Branch Block"

_jcm, 2022, doi:10.3390/jcm11195580_

Round 1
Reviewer 1 Report
The authors presented their retrospective cohort study of patients who underwent TAVR and had RBBB. They intended to look for patients' specific electrophysiological significant findings including novel ECG-, rhythm- or anatomical predictors for the identification of patients with underlying RBBB who are at high risk for PPI or present an increased 1-year cardiac mortality if no PPI was performed after TAVI. They identified 98 patients with pacemaker naive RBBB out of the whole 2129 TAVI cohort (4.2%) which is comparable with other series (Shresta B, et al. Curr Probl Cardiol 2022;47(10).101298). The results of their study showed 43.9% of RBBB patients required PPI. However, their findings did not support their clinical suspicion or support their "patient-specific" approach to predicting post-TAVI pacemaker implantation. It could be when they are analyzing the subgroup RBBB morphology, the sample size is too small to have a clear statistical association with the outcome.
Can they also explain a bit more, why they think different morphology of RBBB should have different risk of PPI? can they provide anatomical and electrophysiological reason for that?
As they have pointed out, it would be very important if they can contact those primary cardiology to interrogate the pacemaker to see how many of these RBBB patients did require trigger.
One of the parameters they want to have a patient-specific evaluation of PPI risk is CT assessment of calcification of the NCC, can they specify who is reading the CT images? independent radiology would enhance the creditability of the findings
Author Response
Dear Reviewer#1
We like to thank you for your thorough review of our paper and for helpful comments. Please find our itemized responses to your questions and suggestions below.
- Can they also explain a bit more, why they think different morphology of RBBB should have different risk of PPI? can they provide anatomical and electrophysiological reason for that?
Thank you for this helpful advice. As conduction disturbances develop in the HIS-Purkinje-system, a different RBBB morphology could have its origin from a different localization within the HIS-Purkinje-system. Conduction disturbances with proximity closer to the HIS-Bundle should lead to a more vulnerable conduction and may be easier affected by an implanted valve. Until now this hypothesis was not yet investigated and there exists no former data. We added this information to the manuscript:
Page 4, paragraph 3, lines34-6:
Conduction disturbances with proximity closer to the HIS-Bundle should lead to a more vulnerable conduction and may be easier affected by an implanted valve and this may be detected by a different RBBB morphology.
Page 4, paragraph 4, lines 4-6:
Our hyptothesis was that RBBB morphology has an influence on PPI rates and higher amount of calcification in the implantation area leads to more PPI.
- As they have pointed out, it would be very important if they can contact those primary cardiology to interrogate the pacemaker to see how many of these RBBB patients did require trigger.
Thank you very much for this comment. There are two main problems, we have to admit:
First, interrogation of pacemaker programming comes with a few problems, that make them difficult to compare: different programming with differently programmed pacing rates, intermittend atrioventricular conduction disturbances and co-medication.
Second, due to European data protection regulation standards, we can not contact cardiologists without patient’s consent. As this analysis is based on retrospective data, we were not able to get these consents from our patients.
Both problems could be addressed by a prospective study design with pre-defined pacemaker-programming and patients’ consent.
- One of the parameters they want to have a patient-specific evaluation of PPI risk is CT assessment of calcification of the NCC, can they specify who is reading the CT images? independent radiology would enhance the creditability of the finding.
Thank you very much.
We added this information to the manuscript:
Page 6, paragraph 3, lines 8-9:
These measurements were performed by two blinded, independent radiologists at our center with no affiliation to this study project.
We hope, that our efforts in improving our manuscript in answering your helpful comments enhance the statements of our manuscript substantially.

Reviewer 2 Report
Schoechlin et al investigate the impact of RBBB on PM rate after TAVI in a retrospective, single center study. They assessed 98 patients. PM rate was 44%. Female patients, male patients treated with balloon-expandable valve types, patients with high load/degree of non-coronary LVOT calcification and patients with atrial fibrillation were identified at risk for PM implantation in patient with RBBB undergoing TAVI.
The manuscript is clearly organized and well written. I have the following comments:
1. Please rephrase the last paragraph of the introduction. What was the aim of the study?
2. On Page 4 the authors state “Indications for PPI in patients with RBBB are displayed in Table 3”. This is not true. Please correct.
3. Page 5 “The overall rate of PPI in patients with RBBB was 43.9% (see indications for PPI, supplemental table 1). Since this is of major relevance, it is important to provide more details. What was the percentage of patients receiving a PM due to high-degree AV block? Sinus bradycardia?
4. What was the time interval until PM implantation? % Day 1, % Day 2, etc.
5. How many patients received a pacemaker before discharge and how many patients received a PM during FU? Did the identified predictors differ between these groups?
6. The authors found that male patients had a significantly lower risk for PPI, when they underwent a balloon-expandable valve type (26.5% vs. 58.3%, p=0.035). What are possible explanations for this finding? Please discuss.
7. In multivariate analysis female sex was the strongest risk factor (4.431, 1.609-12.203; p= 0.004). The authors discuss that a possible reason could be a relevant oversizing of valves, as the aortic annulus in female individuals tends to be smaller in respect to their body size. Why should this only be relevant in patients with RBBB? Please discuss.
8. Similarly, women have lower PM rates after TAVI. This is consistent in the literature. This seems to be different in patients with RBBB. Please discuss.
9. One of the main limitations of this paper which limits the applicability of this results and should be noted in text, is lack of a regression model combining all the risk factors identified (sex, calcification quartile, type of valve, valve size but also baseline PQ interval and QRS duration, age etc.). Without it, we wouldn’t know which of these are independent risk factors. As of such, authors should either include a multivariable regression model using just the risk factors indentified in the paper, or to mention the lack of such model in the study limitations.
10. Table 2 - please include data regarding the valve sizes . This data is improperly reported in the discussions section - page 7, section 4.2., without any mention in the results or in the table.
Minor comments
- Please include the median follow-up time and IQR. Please provide the number of patients lost to follow-up
- Page 5/10 section 3.2. Authors state that “in logistic regression analysis, HR (95%CI) for PPI …“, they most probably refer to a univariable regression model. Please modify accordingly.
- Page 5/10 section 3.2. In the multivariable regression model reported later, it is not clear which variables have been included (only sex and self-expandable valve type?). Please provide a forrest plot and Hazard Ratios and CI of all the variables.
- Page 7/10 section 4.0. - The authors note that “Patients with pre-existing RBBB and atrial fibrillation, who are not treated with PPI have an increased mortality risk of 35.7% at one-year following TAVI.” What what the cause of death? Could these be due to conduction disorders? Patients with AF have generally a higher risk for PM implantation after TAVI.
- Regarding FIgure 1 and Table 3. the first ECG is falsely lableled as a rsR’ complex, when the figure shows a qR complex. Please either change the ECG figure, or the label accordingly. The same goes for last ECG figure, which is not Rsr’ but rather a fragmented QRS complex. A Label “Other” instead of “Rsr’” would be more appropriate. Please do not forget the change the labels in the table as well.
- Figure 2 - please add the log-rank test inside the figure.
Author Response
Title: Risk Stratification for Pacemaker Implantation after Transcatheter Aortic Valve Implantation in Patients with Right Bundle Branch Block
Dear Reviewer#2
We like to thank you for your thorough review of our paper and for helpful comments. Please find our itemized responses to your questions and suggestions below.
- Please rephrase the last paragraph of the introduction. What was the aim of the study?
Thank you for this helpful comment. We clarified our hypothesis by adding the following information to the manuscript:
Page 2, paragraph 4, lines 5-6:
Our hyptothesis was that RBBB morphology has an influence on PPI rates and higher amount of calcification in the implantation area leads to more PPI.
- On Page 4 the authors state “Indications for PPI in patients with RBBB are displayed in Table 3”. This is not true. Please correct.
We changed the wrong link in the text:
Page 4, paragraph 3, line 3:
Indications for PPI in patients with RBBB are displayed in supplemental table 1.
- Page 5 “The overall rate of PPI in patients with RBBB was 43.9% (see indications for PPI, supplemental table 1). Since this is of major relevance, it is important to provide more details. What was the percentage of patients receiving a PM due to high-degree AV block? Sinus bradycardia?
Thank you very much for this great comment. We added the information on PPI indications to the manuscript:
Page 5, paragraph 6, lines 2-4:
The vast majority of PPI were indicated because of third degree AV block (79.1%), followed by symptomatic bradycardia (14.0%), alternating bundle branch block (4.6%), and high-degree (2.3%).
- What was the time interval until PM implantation? % Day 1, % Day 2, etc.
We added this information to the manuscript:
Page 5, paragraph 6, lines 4-6:
Time of PPI was mainly on day of the TAVI-procedure and the day after TAVI-procedure (day 0 = 55.8 %, day 1 = 27.9 %, day 2 = 4.7 %, later than day 3 = 11.6%).
- How many patients received a pacemaker before discharge and how many patients received a PM during FU? Did the identified predictors differ between these group.
We added this information to the manuscript:
Page 5, paragraph 6, lines 1-3:
The overall rate of PPI in patients with RBBB was 43.9% until discharge (see indications for PPI, supplemental table 1), one more patient reveived pacemaker after discharge on day 35 after procedure.
- The authors found that male patients had a significantly lower risk for PPI, when they underwent a balloon-expandable valve type (26.5% vs. 58.3%, p=0.035). What are possible explanations for this finding? Please discuss.
Thank you very much for this substantial comment. We discussed this issue at our research team and came to the result, that the main driving factor was male sex, as female sex was the strongest risk factor in multivariate analysis, as described in our manuscript. We therefore added this information to our manuscript, in the discussion.
Page 7, paragraph 3, lines 4-5:
An interesting finding was that male patients, who received self-expandable valves are nearly at the same increased risk for PPI as female patients, in whom the choice of valve type did not affect their risk for PPI. The reason for the higher number of PPI in female patients is unclear, and we think, that the main factor for our finding in male patients, treated with balloon-expandable valve types, is the male sex.
- In multivariate analysis female sex was the strongest risk factor (4.431, 1.609-12.203; p= 0.004). The authors discuss that a possible reason could be a relevant oversizing of valves, as the aortic annulus in female individuals tends to be smaller in respect to their body size. Why should this only be relevant in patients with RBBB? Please discuss
Thank you for this helpful comment, we discussed this issue in our manuscript:
Page 7, paragraph 3, lines 7-8:
Oversizing is a known and relevant risk factor for PPI19 and could be also an independent risk factor in patients with RBBB.
- Similarly, women have lower PM rates after TAVI. This is consistent in the literature. This seems to be different in patients with RBBB. Please discuss.
Thank you very much for this very helpful comment. We have to indicate, that the reason for this issue remains unclear. We discussed this issue in our research team and postulated that different ways of the female organism in inflammation or hormone levels with influence on the conduction system may be a reason. As these ideas are very speculative, we decided to waive this discussion in our manuscript.
- One of the main limitations of this paper which limits the applicability of this results and should be noted in text, is lack of a regression model combining all the risk factors identified (sex, calcification quartile, type of valve, valve size but also baseline PQ interval and QRS duration, age etc.). Without it, we wouldn’t know which of these are independentrisk factors. As of such, authors should either include a multivariable regression model using just the risk factors indentified in the paper, or to mention the lack of such model in the study limitations.
Thank you very much for this advice. All variables at significance levels p < 0.1 were included in our regression model. We should have stated this statistical proceeding in our methods section:
Page 5, paragraph 2, lines 3-4:
For multivariate regession analysis, all variables with statistical significance levels at p < 0.1 were included.
- Table 2 - please include data regarding the valve sizes . This data is improperly reported in the discussions section - page 7, section 4.2., without any mention in the results or in the table.
We added this information to the manuscript in table 2 on page 3:
Table 2. Procedural characteristics of patients with pre-existing RBBB |
||||
|
All Patients (n=98) |
PPI after TAVI (n=43) |
No PPI after TAVI (n=55) |
P Value |
Dilatation |
|
|
|
|
Predilation |
18 (18.4) |
6 (14) |
12 (21.8) |
0.318 |
Postdilatation |
19 (19.4) |
8 (18.6) |
11 (20) |
0.862 |
|
|
|
|
|
Valve Type |
|
|
|
0.060 |
Self-expandable |
23 (23.5) |
14 (32.6) |
9 (16.4) |
|
Balloon-expandable |
75 (76.5) |
29 (67.4) |
46 (83.6) |
|
|
|
|
|
|
Valve Size |
|
|
|
0.738 |
23mm |
24 (24.5) |
10 (23.2) |
14 (25.5) |
|
26mm |
46 (46.9) |
20 (46.5) |
26 (47.3) |
|
29mm |
28 (28.6) |
13 (30.2) |
15 (27.3) |
|
|
|
|
|
|
Valve in Valve |
3 (3.1) |
- |
3 (5.5) |
0.120 |
|
|
|
|
|
Values are mean ± or n (%) PPI=permanent pacemaker implantation |
Minor comments
- Please include the median follow-up time and IQR. Please provide the number of patients lost to follow-up
We added this information to the manuscript:
Page 3, paragraph 1, lines 3-4:
We report a complete follow-up for this cohort over 1 year, with no patients lost in follow-up.
- Page 5/10 section 3.2. Authors state that “in logistic regression analysis, HR (95%CI) for PPI …“, they most probably refer to a univariable regression model. Please modify accordingly.
We modified this information in the manuscript:
Page 6, paragraph 2, line 1:
In the univariable regression analysis, HR (95% CI) for PPI in patients with RBBB was 0.297 (0.127-0.697) for male.
- Page 5/10 section 3.2. In the multivariable regression model reported later, it is not clear which variables have been included (only sex and self-expandable valve type?). Please provide a forrest plot and Hazard Ratios and CI of all the variables.
As described previously, all variables at signifance levels p < 0.1 were included below multivariable regression models. We did not display further analysis such as forrest plots in our manuscript, as we stated in our limitations section the main limitation of low patient number. This limits the multivariable regression models and we think, that displaying forrest plot would distract from our main findings.
- Page 7/10 section 4.0. - The authors note that “Patients with pre-existing RBBB and atrial fibrillation, who are not treated with PPI have an increased mortality risk of 35.7% at one-year following TAVI.” What what the cause of death? Could these be due to conduction disorders? Patients with AF have generally a higher risk for PM implantation after TAVI.
We added this information to the manuscript:
Page 7, paragraph 1, line 9:
As autopsies were not performed, we do not know causes of death.
- Regarding FIgure 1 and Table 3. the first ECG is falsely lableled as a rsR’ complex, when the figure shows a qR complex. Please either change the ECG figure, or the label accordingly. The same goes for last ECG figure, which is not Rsr’ but rather a fragmented QRS complex. A Label “Other” instead of “Rsr’” would be more appropriate. Please do not forget the change the labels in the table as well.
Thank you very much for your comment. As we used patient ECG data from our IT-System in this figure, we changed this figure to a more “artificial” displaying of ECG sample:
- Figure 2 - please add the log-rank test inside the figure.
We changed appearance of figure 2 according your suggestion:
We hope, that our efforts in improving our manuscript in answering your helpful comments enhance the statements of our manuscript substantially.

Round 2
Reviewer 1 Report
The authors addressed my concerns and I have no further comments.
I endorse the manuscript for publication
Author Response
Title: Risk Stratification for Pacemaker Implantation after Transcatheter Aortic Valve Implantation in Patients with Right Bundle Branch Block
Dear Reviewer#1
We like to thank you for endorsement for publication of our manuscript. Your comments helped us a lot to improve our manuscript.

Reviewer 2 Report
All my comments have been adressed.
Author Response
Title: Risk Stratification for Pacemaker Implantation after Transcatheter Aortic Valve Implantation in Patients with Right Bundle Branch Block
Dear Reviewer#2
We like to thank you for endorsement for publication of our manuscript. Your comments helped us a lot to improve our manuscript.
